# Decoding FGF/FGFR Signaling: Insights into Biological Functions and Disease Relevance

**DOI:** 10.3390/biom14121622

**Published:** 2024-12-18

**Authors:** Oshadi Edirisinghe, Gaëtane Ternier, Zeina Alraawi, Thallapuranam Krishnaswamy Suresh Kumar

**Affiliations:** 1Cell and Molecular Biology Program, University of Arkansas, Fayetteville, AR 72701, USA; ocediris@uark.edu; 2Department of Chemistry and Biochemistry, University of Arkansas, Fayetteville, AR 72701, USA; gternier@uark.edu (G.T.); ziibrahe@uark.edu (Z.A.)

**Keywords:** Fibroblast Growth Factors, FGF, FGF receptor, FGF signaling, mitogens, FGF pathology

## Abstract

Fibroblast Growth Factors (FGFs) and their cognate receptors, FGFRs, play pivotal roles in a plethora of biological processes, including cell proliferation, differentiation, tissue repair, and metabolic homeostasis. This review provides a comprehensive overview of FGF-FGFR signaling pathways while highlighting their complex regulatory mechanisms and interconnections with other signaling networks. Further, we briefly discuss the FGFs involvement in developmental, metabolic, and housekeeping functions. By complementing current knowledge and emerging research, this review aims to enhance the understanding of FGF-FGFR-mediated signaling and its implications for health and disease, which will be crucial for therapeutic development against FGF-related pathological conditions.

## 1. Introduction

Growth factors exhibit essential roles in intercommunication between cells via ligand-receptor cell signaling. A typical cell signaling pathway consists of three clear modes of succession—reception of the signal; transduction through relay molecules; and cellular response by the target cell. The intercommunication between cells is induced when an extracellular ligand binds to a specific receptor in an extracellular compartment of the target cell. This reception is detected by a ligand that induces the transduction signal, a chemical input that acts inside the cell [1,2].

The fibroblast growth factor (FGF) family consists of 22 proteins in humans [3]. Mitogenic fibroblast growth factors are categorized into FGF1, FGF4, FGF7, FGF8, and FGF9 subfamilies [4]. These fibroblast growth factors display various functions such as mitogenic, angiogenic, differentiation, chemotactic, and anti-apoptotic activities. Mitogenic growth factors act on stem cells and various cell types of mesenchymal and epithelial origins. Therefore, mitogenic growth factors improve wound healing and tissue repair [3]. FGFs possess a molecular weight of around 17 to 34 kDa in vertebrates. They exert their metabolic functions via binding and activating specific tyrosine kinase fibroblast growth factor receptors (FGFRs) [1]. Mitogenic growth factors act locally via interaction and activation of cell surface tyrosine kinase FGF receptors (FGFRs) through a high-affinity interaction with heparin or heparan sulfate (HS) [1]. Heparin-dependent FGF-FGFR binding causes receptor dimerization and trans-autophosphorylation of the tyrosine receptors. Activated FGFR triggers the main downstream intracellular signaling pathways (mitogen-activated protein kinase (MAPK), phosphatidylinositol-3-kinases/serine/threonine kinase (PI3K/AKT), extracellular signal-regulated kinase 1/2 (ERK1/2) pathway [5,6], and signal transducer and activator of transcription (STAT)) [3]. Mitogenic FGF signaling mediates various biological and pathophysiological processes, including angiogenesis, wound healing, embryonic development, and metabolic regulation [3] (Figure 1a).

Metabolic FGFs consist of FGF15/19, FGF21, and FGF23. Similar to mitogenic FGFs, metabolic FGFs also drive their functions through the activation of signaling pathways such as MAPK, PI3K–AKT, PLCγ, and STAT. These metabolic FGFs regulate metabolic pathways, and their effect results in metabolic homeostasis of bile acids, lipids, glucose, energy, and minerals [1,5] (Figure 1b).

The FGF-mediated signaling is context-dependent, i.e., depending on the tissue type, local concentration, FGFR isoform, or presence of other signaling molecules, which can ensure different pathways with the same FGF, highlighting the complexity and diversity of FGF signaling [7,8,9]. In this review, we have summarized recent knowledge on FGF-FGFR-mediated cell signaling pathways and the diseases that arise from dysregulation of FGF signaling by a comprehensive literature search using the Web of Science data base (https://www.webofscience.com accessed from 1 August 2024 to 1 November 2024).

## 2. Fibroblast Growth Factor Receptors (FGFRs)

FGFs bind to their receptors (FGFRs) to exert their various biological functions. There are four main types of FGFRs, namely, FGFR1, FGFR2, FGFR3, and FGFR4 [10]. These are composed of three extracellular immunoglobulin domains (D1-D3), a single-pass transmembrane domain, and a cytoplasmic tyrosine kinase domain [1,11]. They possess an acid box that includes an acidic, serine-rich sequence in the linker between the D1 and D2 domains, where the D1 domain controls receptor autoinhibition and the base of the D2 domain assists in the binding of each FGF to the FGF receptors on the cell surface [11]. Each FGF molecule binds to the D2 domain of a second receptor via a secondary receptor binding site. The D2-D3 fragment is essential for ligand binding and specificity [3]. An exon-skipping mechanism removes the D1 domain, giving rise to various isoforms of the receptor (FGFR1-FGFR3), while an alternative splicing process in the second half of the D3 domain of FGFR1–3 forms b (FGFR1b–3b) and c (FGFR1c–3c) isoforms that have distinct FGF binding specificities and are primarily act epithelial and mesenchymal tissues, respectively [1,4]. Heparin sulfate acts as the coreceptor, which facilitates binding of the FGF ligand to FGFRs, initiating dimerization and induction of the cytoplasmic kinase domains, in turn trans phosphorylating and activating the A-loop tyrosine. The two important intracellular substrates of FGFR are PLCγ substrate 1 (also known as FRS1) and FGFR substrate 2 (also known as FRS2) [12]. Phosphorylation of an FGFR-invariant tyrosine (Y766 in FGFR1) at the C tail of FGFR creates a binding site for the SH2 domain of PLCγ and is needed for PLCγ phosphorylation and activation, whereas FRS2 associates constitutively with the juxta membrane region of the FGFR. Phosphorylation of FRS2 is critical for the activation of the Ras-MAPK and PI3K-AKT signaling pathways, hence stimulating downstream signaling pathways [12].

## 3. Fibroblast Growth Factors (FGFs)

### 3.1. Paracrine FGFs

#### 3.1.1. FGF1 Subfamily

The FGF1 subfamily consists of the prototypical FGFR ligands, namely, FGF1 and FGF2. These members are mitogenic FGFs that can stimulate cell proliferation, differentiation, and angiogenesis by acting in a paracrine fashion. Their FGF-FGFR interactions are heparin sulfate dependent [13].

##### FGF1

The crystal structure of FGF1 was first published in 1991 by Zhu et al. (1991) (PDB ID—1BAR) [14] displaying FGF β-trefoil core; characterized by three copies of a four-stranded β sheet and amino and carboxy termini extending outward from the core. Currently, 102 structures of different FGF1-FGFR complexes can be found in the Research Collaboratory for Structural Bioinformatics Protein Data Bank (RCSB PDB) (www.rcsb.org accessed on 20 September 2024). FGF1 is also known as the universal ligand of FGFR due to its ability to bind to all receptor isoforms [15]. The promiscuous nature of FGF1-FGFR interactions was initially postulated to be derived from the high flexibility of the N-terminus of FGF1, which makes fewer contacts between the βC′-βE loop of the D3 regions of FGFR compared to other FGF-FGFR interactions. However, later studies reported that such promiscuity arises due to the ability to participate in diverse interactions of the N-terminus of FGF1 with “b” or “c” isoforms of FGFRs. The nine amino acid residues adjacent to the β-trefoil core (Phe16 to Lys24) are found to be critical for such promiscuous interactions [15].

The mature peptide contains 155 amino acid residues (17–18 kDa). FGF1 lacks a cleavable signaling peptide; hence, it follows a noncanonical secretory mechanism that involves direct translocation into the cytoplasm upon stress (extracellular FGF1) [16]. This is mediated by the formation of many protein complexes. Some important constituents of such multiprotein complexes are S100A13, the p40 form of synaptotagmin 1 [17], and AHNAK2 [18]. Moreover, FGF1 can also be found in the nuclei of cells, suggesting a mechanism of nuclear localization (intracellular FGF1). This was attributed to 5′-nuclear localization signal and alternative translation initiation sites [19]. Nucleolin and phosphorylation of FGF1 by PKCδ were reported to play a critical role in the nuclear export of FGF1 [20].

The classical effects of FGF1 include stimulation of cell proliferation, differentiation, angiogenesis [13], and antiapoptotic activity [21,22]. Hence, FGF1 plays a seminal role during embryogenesis, particularly during the embryonic nervous system, kidney, lung, and limb generation [13]. Moreover, a plethora of research has confirmed the affiliation of FGF1 during wound repair [23]. Further, FGF1 exhibits antiapoptotic properties and facilitates cell survival [21,22]. The maintenance of retinal pigmented epithelial cells is critical for neural function in the retina, and extensive apoptosis of these cells can lead to degeneration of the retina. Together with FGF2, FGF1 promoted retinal pigmented epithelial cell survival by activating FGF1-FGFR1-ERK2 signaling and Bcl-x production [24].

More recently, Adipose-derived FGF1 was reported to be able to exert metabolic effects such as promoting insulin sensitization, lipolysis, and adipose remodeling and stimulating glucose uptake [25] in addition to their well-established mitotic functions. Brown et al. (2020) reported that the antidiabetic effect of FGF1 was regulated via prolonged activation of MAPK/ERK signaling in the hypothalamus, providing more insight into FGF1-mediated metabolic regulation [26]. Jonker et al. (2012) reported that FGF1-mediated adipose remodeling in mice was regulated via PPARγ transcription [27]. Moreover, these metabolic effects of FGF1 are predominantly mediated through FGF1-FGFR1 interactions [25]. By fine-tuning FGF1-FGFR-HS ternary complex stability, the shift from mitotic to metabolic actions can be attained. Hence, engineered FGF1 mutants with reduced heparin sulfate binding will exert reduced mitogenicity with enhanced metabolic potential [28]. Huang et al. (2017) generated an engineered FGF1 mutant (FGF1^ΔHBS^) with reduced heparin binding to test its efficacy against cardiac ischemia-reperfusion. The mutant successfully restored cardiac function via FGF1-FGFR1c-ERK1/2 signaling [28,29]. Interestingly, the same mutant gave rise to the PI3K/AKT pathway and GSK-3β/Nrf2 signaling while inhibiting ASK1/JNK signaling in podocytes in a mouse model for diabetic nephropathy, suggesting the complexity of FGF-FGFR-mediated signaling in different cell types and biological contexts [30].

Overexpression of FGF1 can aggravate certain cancers such as ovarian cancer [31], non-small cell lung cancer [32], colorectal cancer [33], and pancreatic cancer [34]. In ovarian cancer, the WNT7A-β-catenin axis can regulate FGF1 expression in cancer cells by interacting with TCF-binding elements of the FGF1 promoter. WNT7A and FGF1 overexpression can induce tumor growth, leading to poor disease outcomes [31]. Further, several studies reported that the FGFR4 isoform is overexpressed in ovarian cancer patients (in advanced stages) and mainly interacts with FGF1. FGF1-FGFR4 initiated multiple signaling pathways, including MAPK, nuclear factor-kappa B (NF-κB), and WNT signaling that promoted tumor cell survival, proliferation, and metastasis [35,36,37]. However, in colorectal cancer, FGF1-FGFR3 signaling promoted cancer metastasis [38] (Table 1). These findings confirm that FGF1 interactions vary depending on the environment and temporal cues.

##### FGF2

FGF2 is also known as the basic fibroblast growth factor due to its high isoelectric point. The mature peptide contains 146 amino acids [39]. Plotnikov et al. (1999) resolved the X-ray crystal structure of FGF2 conjugated with FGFR1. The FGF2 protein had the characteristic β-trefoil core and was found to interact with D2 and D3 regions of FGFR1. Further, a cluster of positive charges in FGF2 suggests its affinity for heparin binding. Moreover, FGF2 shows a higher affinity for FGFR1c and FGFR2 than FGFR3c and FGFR4. The differences in lengths of βC′-βE loop of FGFR determine the aforementioned receptor preference of FGF2 (the FGFR4 βC′-βE loop is shorter by two amino acid residues than FGFR1-3c, and FGFR3 has an alanine residue at 317th position instead of valine in FGFR1c, impairing the optimal interactions with FGF2) [39].

FGF2 does not contain a signal peptide; hence, it cannot be secreted via the classical secretion pathway mediated by ER/Golgi. In recent literature, FGF2 was reported to be secreted via directly interacting with Na^+^/K^+^-ATPase α1, facilitating the recruitment of the protein into the cell in an unconventional route. Further, phosphorylation by Tec-kinase and competition between heparin sulfate and 4,5-bisphosphate for membrane translocation of FGF2 play critical roles as well [40]. FGF2 is associated with cell proliferation, differentiation, and angiogenesis [41,42]. Moreover, it plays pivotal roles during osteogenic differentiation [43,44,45], neural development [7], kidney development [46], oocyte maturation [47], and spermatogenesis [48]. Further, FGF2 was reported to confer protection against anxiety [49,50], enhance remyelination [51], improve myocardial perfusion [52] and lens regeneration in eyes [53], and prevent cardiotoxicity induced by sunitinib [54]. However, in several types of cancer, FGF2 expression was elevated (i.e., colorectal cancer, gastric cancer, ovarian cancer, and pancreatic cancer [55]). Furthermore, dysregulated FGF2-FGFR signaling can result in osteoarthritis [56], autoimmune arthritis [57], blinding eye disorders (e.g., age-related macular degeneration [58]), and puromycin aminonucleoside nephropathy [59].

Angiogenesis is the process of the formation of new blood vessels, and FGF2 is reported to be a key driver during the process. FGF2-mediated angiogenesis is critical for embryonic development, tissue repair, and wound healing [60]. During angiogenesis, FGF2 interacts with FGFR1c in endothelial cells and recruits an adapter protein, FRS2α, giving rise to downstream MAPK/ERK signaling. Further, vascular endothelial growth factor (VEGF) activation has also been reported to play a critical role. Zhu et al. (2022) reported that small ubiquitin-like modifier modification (SUMOylation; a reversible posttranslational modification on lysine residues) by SENP1 on FGFR1 inhibits/downregulates the receptor tyrosine kinase activity of FGFR1 while facilitating recruitment of FRS2α to the VEGF receptor complex during hypoxic conditions [60]. The study further mentioned that the competitive recruitment of FRS2α to either FGF2-FGFR1c or VEGF-VEGFR is the key mediator in the determination of the predominant pathway of angiogenesis since in normoxic conditions, FGF2-FGFR1 signaling activity predominates, whereas in hypoxic conditions VEGF-VEGFR signaling is the main pathway of angiogenesis, suggesting a complex mechanism of regulation of vascularization [60]. Hence, FGF2-FGFR1c-MAPK-ERK signaling is critical during wound healing [60,61]. However, during pathological conditions such as cancer, FGF2 is reported to interact with FGFR3 to promote angiogenesis. Ren et al. (2014) reported that, in osteosarcoma, FGF2-FGFR3 interacts with apurinic/apyrimidinic endonuclease 1, promoting tumor vasculature [62].

The dysregulated neovascularization observed in blinding diseases was reported to be associated with FGF2-FGFR-STAT3 signaling [58] and, moreover, cidofovir, an antiviral drug targeted for cancers due to human papilloma viral infections, enhanced apoptosis of tumor cells and attenuated FGF2-driven progression of vascular tumors through FGF2-FGFR-mediated Erk42/44 activation in endothelial cells [63]. During osteogenic differentiation, FGF2 activated TAZ expression and facilitated nuclear localization. In the nucleus, together with Runx2, TAZ activated osteoblast differentiation. This was further stimulated by FGF2-mediated ERK signaling [43]. Moreover, during nephrogenesis, LIF and TGFβ2 transcription factors acted in concert with FGF2 to coordinate the formation of rat renal tubules via Wnt signaling [46]. Lastly, the PLCγ/IP3/Ca^2+^ signaling axis (during endothelial cell metabolism and angiogenesis) [64] and FGF2-MAPK2K1/AKT signaling axis (in spermatogonial cell proliferation) were reported to play essential roles [48].

#### 3.1.2. FGF7 Subfamily

FGF3, FGF7, FGF10, and FGF22 are categorized into the FGF7 subfamily based on their similarities in primary amino acid sequence. The FGF-FGFR interactions are heparin-dependent. The members are mesenchymally expressed, and their biological functions are mediated solely by the “b” isoforms of FGFRs [65]. The exclusive preference of FGF7 members for the “b” isoform of FGFRs is determined by the specific contacts that they create with D3 (the alternatively spliced region) and D2 domains of FGFRs. Further, variable lengths in amino and carboxy termini, differential HS binding affinities, and variable strengths in FGFR dimerization potentials might accentuate biological functions unique to each member of this subfamily [65].

##### FGF3

FGF3 is encoded by 11q13.3 [66]. The molecular structure of FGF3 has not been discovered in the published literature thus far. However, the FGF3 structure is postulated to adopt the characteristic β-trefoil fold based on sequence homology [65]. Interestingly, FGF3 is known to have two different subcellular fates: a secreted form that induces a mitogenic effect and a nuclear-localized form that inhibits cell proliferation [67,68]. FGF3, derived from the 5′AUG codon (239 amino acid residues in humans [69]), is secreted and acts in a paracrine manner during embryogenesis and cancer pathology. Conversely, FGF3 transcripts translated from upstream to an in-frame CUG codon (with an additional 78 nucleotides in humans that translated into a 273 amino acid residue long peptide [69]) harbor dual subcellular fates. Similar proportions of CUG-derived FGF3 are secreted or directed into the nucleus. The subcellular destination of FGF3 is determined by N-terminal signaling sequences [67], basic regions in the carboxy terminus [70], and trafficking pathways [68].

The importance of FGF3 in ear development is well-studied in the literature. Maroon et al. (2002) reported that FGF3 and FGF8 were used during the otic placode, where they stipulated that the FGF3 and FGF8 combinatory effects are critical at the initial stages of the otic placode [71]. These findings were further confirmed by Liu et al. (2003), who found that sox9a expression is dependent on FGF3 and FGF8 for the epithelial organization during otic placode development [72]. Further, FGF3 and FGF8-mediated inner ear development is regulated in a spatiotemporal manner via Wnt8 expressed by the hindbrain during otic induction [73] and inter-rhombomere signaling [74]. Hatch et al. (2007) reported more detailed information on the aforementioned WNT and rhombomere-mediated signaling, where WNT signaling induced otic patterning dorsally and FGF3-rhombomere signaling prevented ventral expansion in mice [75].

The involvement of FGF3 in brain development is also reported. Reuter et al. (2019) reported that FGF3 is directly involved in hypothalamus development by regulating the development of populations of serotonergic cerebrospinal fluid-containing cells, dopaminergic, and neuroendocrine cells [76]. FGF3 plays a critical role in the accurate segmental development of the hindbrain via inhibiting BMP activity in chickens [77]. In zebrafish, FGF3 is necessary for forebrain development [78]. However, FGF3 is reported as not essential for mouse forebrain development, underscoring the species-specific effects of FGF3 in brain development [79].

Navigating/pathfinding of chick thalamocortical axons was reported to be mediated by FGF3-FGFR3 interactions [80], underscoring the chemorepellent effect of FGF3 during axon pathfinding. FGF3 regulates vertebrate axis extension and termination, the closing of the neural tube, and the specification of the neural crest during embryogenesis via BMP signaling [81].

Moreover, FGF3 is associated with the onset of different types of cancer, such as human Kaposi’s sarcoma [82], hepatocellular carcinoma [83], breast cancer [84], Laryngeal squamous cell carcinoma [85], and nonsmall cell lung carcinoma [86]. In breast cancer cells, FGF3 was found to be overexpressed, which led to overstimulation of FGF3-FGFR1-STAT3 signaling [84].

##### FGF7

FGF7 is also known as the keratinocyte growth factor, encoded on chromosome 15 q 15-21.1. The mature peptide has 194 amino acid residues. Like other members in this subfamily, FGF7 also acts in a paracrine manner solely via FGFR2b receptor-HS interactions. Ye et al. (2001) published the crystal structure of FGF7 (PDB ID—1QQL) [87]. The crystal structure revealed that FGF7 also contains the β-trefoil structure; however, unlike FGF1 and FGF2, only 10 out of 12 β-strands were antiparallelly oriented in an ordered manner. However, FGF7 still has the FGF-characteristic trifold symmetry. The lengths of some β strands deviated from FGF1 to FGF2 (i.e., β-strand 1 has 3 additional amino acids, and β-strand 6 and 7 of FGF7 are only three amino acid residues long, much shorter than FGF1 and 2). The Cys17 and Cys83 in FGF7 are conserved and do not participate in disulfide bonding similar to FGF1 and FGF2. The heparin-binding domain of FGF7 is also conserved and comprised of Arg18, Asn92, Ans114, Gln115, Val120, Lys124, Gln129, Lys130, and Thr131. The Cα backbone of FGF7 is similar to FGF1 and FGF2; however, the distribution of surface electrostatic potential is different. Contrary to concentrated positive surface potential in FGF1 and FGF2, FGF7 shows a much-dispersed positive charge distribution. Further, the presence of FGF7-specific Glu128 in the proximity of its heparin-binding domain can decrease heparin binding [87]. Wang et al. (1995) reported that His314Thr and Ser315Ala in FGFR2b (βC′-(βE loop) can completely abrogate FGF3-FGFR2b binding [88]. Further, Ron et al. (1993) reported that truncation of the FGF3 N-terminus (from 1 to 63 to 1 to 65 amino acids) led to the complete abolishment of FGF3-mediated cell proliferation in vitro [89]. Moreover, Val103 [90] and Leu142 [91] are reported to be critical in the FGF7-FGFR2b interaction.

The association of FGF7 in bone development has been reported in the literature. Jeon et al. (2013) reported that the exogenous addition of FGF7 to mouse embryonic stem cells promoted osteogenic differentiation (characterized by the formation of bone-like nodules, accumulation of calcium, and bone-specific gene expression) via the ERK-Runx2 pathway. However, FGF7 did not show any effects on embryonic stem cell proliferation [92]. Conversely, bone marrow-derived mesenchymal stem cells elicited osteogenic differentiation via the FGF7-MEK-ERK pathway under direct regulation of MiR-381-3p. MiR-381-3p was reported as a negative modulator of FGF7 expression [93], suggesting tissue-specific differential regulation. FGF7 is also involved in bone formation and homeostasis. FGF7 enhanced dendrite elongation (by regulating E11) in osteoblasts and the formation of gap junctions (by regulating connexin 43) via MAPK and PI3K-AKT pathways [94]. FGF7 is also involved in mammary gland morphogenesis (through the FGF7-TGFα-MAPK-ERK1/2 pathway) [95], cartilage development [96], lung development in rats (by inducing differentiation of alveolar type II cells) [97], vertebrate limb development in chick embryos (by directly inducing apical ectodermal ridge development) [98], decidualization of endometrium in humans (via the FGF7-ERK-JNK pathway) [99], bovine oocyte growth (via FGF7-KIT/KITLG signaling) [100], development of pancreas (via the MAPK pathway in pancreatic progenitor cells) [101], and branching morphogenesis of submandibular glands in mice (via a complex network of FGF7-FGFR2b and FGFR1b-FGF1-MMP2) [102].

Despite its beneficial involvement in organogenesis and development, several cancers arise due to FGF7 overexpression. Feng et al. (2024) recently published that cancer-associated fibroblasts can express FGF7, leading to a higher concentration of FGF7 in the tumor microenvironment during ovarian cancer. Higher levels of FGF7 were associated with increased metastatic potential and poor prognosis. This was postulated to be associated with FGF7-FGFR2b-mediated inhibition of hypoxia-inducible factor 1 alpha degradation [103]. In breast cancer patients, FGF7 was overexpressed and showed a significant association with tumor cell proliferation, angiogenesis, and epithelial-to-mesenchymal transition in mammary cells [104]. Other than that, higher FGF7 levels are associated with ameloblastoma proliferation (via MAPK pathway) [105], retinoblastoma invasion [106], cholangiocarcinoma progression, and pemigatinib resistance (via inducing Wnt-TCF7-SOX9 signaling through FGF2-FGFR2b interaction) [107], leukemia/myeloblastoma [108], and urothelial carcinoma [109].

Further, Shaoul et al. (2006) reported that elevated levels of FGF7 were associated with gastric inflammation, gastric adenocarcinoma, and gastric cancer [110]. Huang et al. (2017) provided much detailed information on FGF7 molecular mechanisms related to gastric cancer pathology. They reported that thrombospondin 1 (which facilitates angiogenesis, tumor growth, and metastasis at the tumor site) expression was upregulated via FGF7-FGFR2b interaction through PI3K/Akt/mTOR signaling. Knocking down thrombospondin 1 inhibited FGF7-induced gastric cancer growth and metastasis [111].

##### FGF10

FGF10 exerts its biological functions predominantly in a paracrine manner via exclusively interacting with FGFR2b. The FGF10 protein is comprised of 208 amino acid residues, where the first 37 residues constitute the signaling peptide, which directs the peptide to the endoplasmic reticulum-Golgi complex for secretion [112]. The crystal structure of FGF was published in 2003 (PDB ID—1NUN) [113]. FGF10 also possesses the characteristic β-trefoil structure with 12 antiparallelly oriented β-strands. Similar to FGF7, the β1 strand of FGF10 is longer, and the β10 and β11 strands are less organized compared to FGF1 and FGF2. The first five amino acid residues at the N terminus of FGF10 are organized into a short helix (gN) and connect to the β1 strand via a short loop. Hence, the N-terminus of FGF10 bound to FGFR2b is more defined than the N-termini of FGFR2c-bound FGF1 and FGF2. FGF10 interacts with the D3 region of FGFR2b by forming an extensive network of contacts supporting higher receptor affinity and specificity of FGF7 subfamily members for the “b” isoform of FGFR2. In FGF10, βB′ strand, βB′-βC loop, N terminus, β1 strand, β4 strands, and β7-β8 loop participated in forming contacts with D3 of FGFR2b. Asp76, Arg78, and Thr114 are some of the critical residues for FGF10-mediated cell proliferation confirmed by mutagenesis experiments [65].

FGF10 is a key regulator in glandular morphogenesis [114,115,116] and lung development [117] during embryonic development. The importance of FGF10-FGFR2b signaling has been widely studied during alveolar development [118]. Moreover, FGF10 function is critical during neural development [119] and tooth development [120]. Hence, mutations in the *Fgf10* gene cause glandular defects (e.g., Lacrimo-auriculo-dento-digital syndrome (LADD) and aplasia of the lacrimal and salivary glands (ALSG)) [121,122,123] and lung defects (e.g., lethal lung developmental disorders [124,125], chronic obstructive pulmonary disease [126], bronchopulmonary dysplasia [118]). Further, several cancers can arise due to *Fgf10* gene mutations or/and FGF10 overexpression, such as prostate cancer, breast cancer, cholangiocarcinoma [127,128], and gastric cancer [118,129].

Further, FGF10 has been recently identified as a molecule conferring cardiac protection, which promotes cardiac regeneration via activating Meis1 expression and enhancing glycolysis to promote cardiomyocyte regeneration and prevent fibrosis [130]. Moreover, FGF10 mediated protection for ischemia-reperfusion injury of the liver in mice via decreasing oxidative stress through the FGF10-FGFR2b- NRF2-PI3K-Akt pathway [131] and attenuated traumatic brain injury in mice [132] and spinal cord injury [133] via modulating the FGF10-FGFR2b-TLR4-MyD88-NF-κB pathway.

##### FGF22

FGF22, encoded on chromosome 19p13 [134]. The mature protein is composed of 170 amino acid residues. The structure of FGF22 has not been published thus far. However, based on similarity in the amino acid sequence, it is assumed to have the characteristic β-trefoil structure. It is reported to act in a paracrine manner [65].

FGF22 is involved in neural development and homeostasis [135], brain development [136], and auditory development [137]. The importance of FGF10 in zebrafish forebrain development was also reported, where FGF10 facilitated forebrain patterning via FGF10-FGFR2b signaling that promoted neurogenesis and gliogenesis [138]. Further FGF22-mediated stimulation of growth plate development of proximal tibia (postnatal) in rats has been reported [139]. Furthermore, FGF22 signaling via FGFR1b and FGFR2b (together with FGF7 and 10) stimulated the proliferation and survival of A6-expressing hepatocytes via the AKT-β-catenin pathway. This pathway was effective against liver injury and promoted liver regeneration in mice [140]. Jacobi et al. (2015) reported that FGF22 can promote spinal cord remodeling upon injury through synapse formation through FGF22-FGFR1b and FGF10-FGFR2b interactions [141]. Further, mice treated with FGF22 showed less ER stress, ER-stress-induced apoptosis of neuronal cells, enhanced neuron abundance, and axon regeneration during recovery from spinal cord injury [141,142]. FGF22-mediated excitatory nerve termini organization was mediated by FGFR1b and FGFR2b via FRS2-PI3K signaling [143], and selective localization of FGF22 to appropriate synaptic locations was mediated by a distinct set of motor (KIF3A and KIF17) and adapter proteins (SAP102) [144].

#### 3.1.3. FGF4 Subfamily

FGF4, FGF5, and FGF6 possess an essential role in various cellular processes, particularly embryonic development and tissue repair. These fibroblast growth factors are involved in cell proliferation, differentiation, and tissue morphogenesis [1]. FGF4 is highly expressed in the ectoderm, mesoderm, and tail bud [145].

##### FGF4

FGF4 shares around 30% sequence identity with the FGF1 and FGF2. Unlike FGF1 and FGF2, FGF4 has a classical signal peptide and thus is efficiently secreted from cells. FGF4 binds and activates the IIIc splice forms of FGFR1 to FGFR3. However, it exhibits little activity towards the b-splice forms of FGFR1 to -3 as well as towards FGFR4 [146]. Like FGF1 and FGF2, FGF4 needs heparin to be biologically active. However, it has been demonstrated that 2-O- and 6-O-desulfated heparin were able to assist the mitogenic activity of FGF4, while neither of these heparins could assist the biological activity of FGF1 and FGF2 [147].

FGF4 plays a prominent role during embryogenesis in multiple ways, such as coordination of the primitive endoderm layer [148,149], limb bud development [150], trophoblast formation [151], notochord formation [152], and neural differentiation [153]. Further, FGF4 is associated with inducing angiogenesis and neuroprotection; hence, it confers protection against cardiac [154,155], neural [156], and liver injury [157]. However, overexpression of FGF4 is reported to be associated with aggravation of tumorigenesis and invasion in many cancers, such as hepatocellular carcinoma [158], lung adenocarcinoma [159], urinary bladder cancer [160], ovarian cancer [161], and breast cancer [162].

In pigs, FGF4-FGFR4-mediated MAPK-ERK signaling resulted in elongation of the trophoblast; however, species-specific variations in downstream signaling cascades were reported [151]. In mice, FGF4 derived from embryonic stem cells stimulated FRS2α-mediated stimulation of ERK signaling. This led to the activation of Cdx2 transcription in the trophoblast stem, which subsequently activated BMP4 signaling and well-regulated development during early embryogenesis [163]. During organogenesis of teeth, lymphoid enhancer factor 1 activates Wnt and FGF4 signaling, which coordinates the communication between epithelium and mesenchyme [164], whereas during limb patterning, SHH-FGF4 signaling is involved [165].

On a positive note, loss of *Fgf4* in mice increased hepatic steatosis and liver damage, suggesting FGF4 is important to maintain hepatic health. Treating mice with liver injury (due to a high-fat diet) with recombinant FGF4 ameliorated liver damage by activating FGF4-FGFR4-Ca^2+^-Ca^2+^/calmodulin mediated AMPK-caspase 6 signaling. Further, the treatment improved fatty acid oxidation while increasing the survival of liver cells [157]. However, in another mouse model with autoimmune hepatitis, FGF4 exerted its protection against liver damage via stimulating CISD3 expression and Nrf2-HO-1 signaling [166]. These findings suggest that FGF4-mediated signaling in the liver is diverse and context-specific.

Moreover, such diversity in signaling pathways can be seen in FGF4-related cancer pathology. In lung adenocarcinoma, overexpression of FGF4 promoted epithelial to mesenchymal transition of tumor cells by elevating store-operated calcium entry and expression of calcium signal-associated protein Orai1 aggravating cancer [159]; however, in breast cancer, the combinatory role of the oncoprotein hepatitis B X-interacting protein (HBXIP) and Sp1 transcription factor activated the FGF4 promoter, giving rise to FGF4 overexpression [162]. Further, during wound healing, FGF4 stimulated keratinocyte proliferation and migration via p38-MAPK signaling and GSK3-mediated Slug stabilization [167].

##### FGF5

FGF5 possesses a highly conserved core region, which consists of 12 β strands and a signal peptide at the N terminus [168]. The mature protein contains 270 amino acid residues. FGF5 has an important role in the regulation of cell proliferation and differentiation and is required for normal regulation of the hair growth cycle. FGF5 inhibits hair elongation by promoting progression from anagen, the growth phase of the hair follicle, into catagen, the apoptosis-induced regression phase. FGF5 is associated with hair length regulation in many species other than humans, such as dogs, cats, sheep, and Syrian hamsters. The *Fgf5* gene-knocked-out sheep displayed longer wool (hair length) with increased active hair follicle density. Crosstalk among many pathways, such as FGF5-FGFR1-mediated Wnt-β catenin signaling, the sonic hedgehog (Shh) pathway, and androgens (i.e., testosterone), can regulate hair growth [169]. Chen et al. (2020) reported an FGF5-mediated protective role after peripheral nerve injury, where Schwann cells reported to overexpress FGF5, which enhanced FGF5-mediated N-cadherin expression while inhibiting ERK 1/2 and MAPK signaling in an autocrine fashion. This promoted Schwann cell adhesion and migration during peripheral nerve regeneration upon injury [170]. Similarly, FGF5 conferred cardiac protection against sepsis and myocardial infarction. Targeted overexpression of FGF5 during sepsis-induced heart injury downregulated CaMKII and NF-κB phosphorylation, reduced oxidative stress, and pyroptosis, improving cardiac injury [171].

However, in osteosarcoma, overexpression of FGF5 resulted in enhanced tumor cell proliferation and metastasis via MAPK signaling [172]. In melanoma, FGF5-MAPK-NFAT signaling resulted in enhanced invasion and clonogenicity, worsening the disease outcome [173], whereas in nasopharyngeal carcinoma, FGF5-FGFR2 interaction ensued Keap1, Nrf2, and HO-1 activation in cancer-associated fibroblasts, leading to decreased sensitivity to cisplatin-based chemotherapy, attenuating its efficacy [174].

##### FGF6

FGF6 interacts with cell-surface-associated heparan sulfate proteoglycans and binds to tyrosine kinase FGF receptors (FGFR1, FGFR2, and FGFR4), with the highest affinity to FGFR4. FGF6 is the only member of the fibroblast growth factor family that is highly expressed in skeletal muscle. FGF6 may not be essential for normal skeletal muscle development in all species, but it possesses a role in myogenesis and muscle regeneration both in early development and adulthood (14) via FGF6-FGFR4 signaling [175].

In *Fgf6^(−/−)^* mice, when injected with higher doses of human recombinant FGF6, it stimulated myogenic stem cell proliferation; however, in mice injected with lower doses, it initiated muscle differentiation by activating calcineurin signaling [176]. Moreover, during skeletal muscle atrophy due to nerve injury, FGF6 exerted a protective effect by enhancing the survival and migration of primary myoblasts. This was coordinated by FGF6-FGFR1-dependent upregulation of cyclin D1, while FGF6-FGFR4 interaction stimulated ERK1/2 signaling that led to upregulation of Myogenin, MyHC, and MyoD expression. Altogether, this led to muscle regeneration [177]. Further, Laziz et al. (2007) reported that muscle regeneration can also be controlled via Sprouty (Spry) proteins. More specifically, in Fgf6 knockout mice, Spry 1,2, and 4 gene expression was elevated in the absence of FGF6, suggesting a complex regulatory network [178]. More recently, the role of FGF6 in cardiac repair has been unfolded. In a mouse model for myocardial infarction, FGF6 expression was found to be upregulated and associated with a reduction in infarction size and better heart functioning. Accumulation of YAP due to FGF6-ERK1/2-mediated inhibition of the Hippo pathway was reported to be effective at stimulating cardiac repair upon myocardial infarction [179]. Han et al. (2012) reported the involvement of FGF6 during tongue development and morphogenesis through TGFβ-Smad4-FGF6 signaling. The pathway specifically promoted myogenic differentiation and fusion of myoblasts during tongue myogenesis [180].

In bladder cancer, overexpression of FGF6 was associated with poor disease outcomes. An oncogene named long intergenic non-protein coding RNA 265 (LINC00265) was reported to be able to bind to miR-4677-3p. This resulted in an elevation of FGF6, suggesting abrogation of negative regulation on FGF6 expression by miR-4677-3p, which facilitated the proliferation and metastasis of cancer cells [181]. In malignant meningioma, overexpression of FGF6 stimulated FGFR-AKT-ERK 1/2 signaling [182].

#### 3.1.4. FGF8 Subfamily

FGF8 subfamily members are FGF8, FGF17, and FGF18. They exert their biological functions in a paracrine fashion, and the signaling is dependent on the heparin sulfate coreceptor. These members are associated with the development of kidney [183], neural [184], cardiovascular [185], cranium, and facial development [185] during embryogenesis.

##### FGF8

Human *Fgf8* is located on chromosome 10q24.32 [186], and its expression in human adults is observed in the kidneys, prostate, breast, bone marrow, testes, and peripheral blood leukocytes [187,188,189]. FGF8 is a mitogenic protein. FGF8 has been studied in depth and based on its balanced role in cell proliferation and differentiation, is a good representation of the fibroblast growth factor family. FGF8 is involved in embryonic development. Studies on mouse embryogenesis revealed that FGF8 is expressed at locations that regulate growth and patterning, such as body axis elongation, midbrain/hindbrain junction, and apical ectodermal ridge (AER) of the limb [190,191]. Further, FGF8 is associated with the development of kidney [183], neural [184], cardiovascular [185], cranium, and facial development [185]. Further, it is associated with glandular formation, such as submandibular salivary glands [192] and mammary glands [193]. However, overexpression of FGF8 in humans occurs in benign breast and prostate lesions, in breast and prostate cancer, and bone metastases in prostate cancer [189,194,195].

In chickens, limb development is mediated via Shh/FGF8 signaling. Further, in mice, FGF8-FGFR2 signaling regulated the limb induction [196]. During the midbrain and hindbrain development of chickens, FGF8-mediated induction of Wnt1 and En1 while repressing Otx2 expression facilitates proper brain patterning [197]. Moreover, during kidney development, FGF8 interactions with WNT8 activated Lim1 expression, leading to enhanced metanephric mesenchymal cell survival and tubulogenesis [183]. In prostate cancer, the FGF8b isoform was reported to be overexpressed in cells expressing androgen receptors (LNCaP cells), aggravating the cancer [198]. In colorectal cancer, YAP1-mediated activation of FGF8 transcription led to overexpression of FGF8 [199]. In breast cancer, HBXIP interacting with CREB can induce FGF8 promoter activity, leading to the initiation of FGF8 transcription and expression. FGF8 can then induce VEGF expression by acting in an autocrine/paracrine manner via the PI3K/Akt/HIF1α signaling axis [200].

##### FGF17

The human FGF17 comprises 2 isoforms, FGF17a and FGF17b. The introns, however, are different in FGF17 and FGF8. Due to the similarities in both gene structure and sequence, it is proposed that the FGF8 and FGF17 genes may have evolved due to evolutionary duplication events. FGF17 plays a major role in mitogenic activities involving cell survival. FGF17 is involved in broad biological processes, including embryonic development, tissue repair, cell growth, tumor growth, and invasion. The expression of FGF17 occurs during embryogenesis and in the cerebellum and cortex of adults, which indicates a key role in neuropsychiatric diseases [201]. Previous studies have also revealed that FGF17 promotes the proliferation and differentiation of oligodendrocyte progenitor cells [202]. Expression has also been observed in prostate epithelial cells, where FGF17 promotes epithelial proliferation by acting as an autocrine growth factor [203].

FGF17 directly interacts with signaling molecules WNT signaling pathway and Gli3 transcription factor during dorsal telencephalon patterning in mice. Together with FGF8 and Wnt-β-Catenin, FGF17 regulates Gli3 expression. Both FGF8 and FGF17 are involved in the enhancement of Wnt8b in the murine cortex. Interestingly, Fgf17 expression was downregulated by Wnt-β-Catenin signaling, restricting FGF17-mediated forebrain patterning. Inferring the optimal balance of these signaling pathways is critical to achieving robustness to mammalian forebrain patterning [204]. Huang et al. (2024) reported that low levels of FGF17 were found to be associated with ischemic stroke (in human subjects) and after blood-brain barrier disruption (in mice). The exogenous treatment of recombinant FGF17 ameliorated blood-brain barrier disruption due to ischemia-reperfusion and reduced endothelial cell apoptosis by activating FGF17-FGFR3-PIK3-AKT signaling, suggesting a potential therapeutic strategy [205].

Despite the aforementioned positive roles in cervical cancer, FGF17 aggravated the disease condition by stimulating perineural invasion. The *ADCYAP1* gene was found to be overexpressed in cervical squamous cell carcinoma and in adenocarcinoma, which promoted the secretion of PACAP from the cancer cells and led to the dedifferentiation of Schwann cells. The dedifferentiated Schwann cell overexpressed FGF17, which subsequently led to the production of Cathepsin S and MMP-12-mediated extracellular matrix degradation, inducing tumor metastasis [206].

##### FGF18

FGF18 is a glycosylated secretory protein with a molecular weight of 20.2 kDa. FGF18 possesses an initial 26 hydrophobic amino acid, which serves as a signal peptide [207]. FGF18 is prominently conserved among humans, mice, and rats (99% identity). As part of the FGF8 subfamily, FGF18 shares a gene structure similar to FGF8 and FGF17 [208]. In general, the function of FGFs is dependent on their receptors, and studies have shown that FGF18 has a high affinity for the FGFR3 receptor and a reduced affinity for FGFR1 [209,210].

FGF18 has a crucial role in major biological processes, including proliferation, migration, motility, and invasiveness of diverse cells; however, FGF18 functions mainly in limb development and skeletal growth by exerting its influence on chondrogenesis and osteogenesis [210]. An in vivo experiment in FGF18 knockout mice reported a lethal implication at the embryonic stage due to delayed ossification, defects in joint development, delayed calvarial suture closure, increased chondrocyte proliferation, and increased hypertrophic zones in the growth plate [208,211,212]. Both in vitro and in vivo studies on mice, rats, and chick embryos have also reported the expression of FGF18 in embryonic development in diverse organs, including the midbrain, lungs, pancreas, muscles, and the intestinal tract [207,208,213,214,215,216]. Additionally, in human heart embryos, FGF18 has also been detected in the cephalic and mandibular mesenchyme [217].

During bone development, Wnt signaling activated transcription factors, namely, TCF and Lef, that subsequently activated the *Fgf18* promoter, resulting in regulated skeletal development. The authors further reported that Runx2 can interact with TCF and Lef in order to activate FGF18 expression [218]. Further, during chondrogenesis, FGF18-FGFR3 signaling promoted cartilage production [219]. Low levels of FGF18 are associated with osteoarthritis pathology, where FGF18-mediated PI3K/AKT signaling is critical to promote chondrocyte proliferation, suppress IL-1β-mediated chondrocyte apoptosis, reduce oxidative stress, and restore mitochondrial function [220].

An increased expression of FGF18 mRNA and protein has been correlated with the upregulation of colon and ovarian tumors [221]. The expression of FGF18 is upregulated through the activation of Wnt pathways, as seen in most colorectal carcinomas [222]. In ovarian cancer, overexpression of FGF18 led to activation of the NF-κB pathway, which was responsible for increased tumor progression, angiogenesis, and metastasis [221].

#### 3.1.5. FGF9 Subfamily

FGF9 subfamily members are FGF9, FGF16, and FGF20. The family exerts a unique regulatory mechanism via reversible self-dimerization to maintain optimal FGF concentration in the body. The dimerized form of FGF9 family members is biologically inactive since amino acid residues that interact with FGFR D2 and D3 regions were reported to be occluded during homodimerization [223]. Hence, only the monomeric form with exposed FGFR D2 and D3 interacting regions exerts biological function in a heparan sulfate-dependent manner [223]. FGF9 members are predominantly paracrine mediators; however, an autocrine mode of action has also been reported. This subfamily plays a key role in male sex determination [224], neural development [225], and organogenesis [226].

##### FGF9

FGF9 is encoded on chromosome 13, and the mature peptide contains 208 amino acids. It predominantly functions in a paracrine fashion via interacting FGFR3c. The crystal structure of FGF9 was first published in 2001 (PDB ID—1IHK) [223]. FGF9 also possesses the β-trefoil core organization; however, the β1-β2 loop of FGF9 is one amino acid residue shorter, and four additional residues are present in the β9-β10 loop (compared to FGF1). Contrary to the FGF1 and FGF2 N- and C-termini, the FGF9 N- and C-terminal are more organized and ordered to form short α helices (αN and αC, respectively.) [223]. The C-terminal of FGF9 is significantly longer than FGF1 and FGF2 (17 extra residues). The FGFR3c preference of FGF9 is inferred to be due to less steric hindrance at the β8-β9 loop of trefoil core and αN region of FGF9 with βC′-βE with FGFR3c during FGF9-FGFR3c interaction compared to FGF9-FGFR1c. For the paracrine actions of FGF9, HS is essential as the coreceptor [223]. One unique feature of FGF9 is its ability to reversibly homodimerize. Formation of FGF9-FGF9 homodimers hinders/buries residues that are crucial to interact with FGFR D3 and D2 regions, suggesting FGF9 homodimers are biologically inactive. Hence, this homodimerization of FGF9 might be a natural regulatory mechanism to maintain the biologically active monomeric FGF9 concentration at the optimal level [223].

More recently, Li et al. (2020) reported that FGF9-mediated male sex determination is regulated by the SRY transcription factor. SRY promoted FGF9 and SOX9 expression from murine testis, leading to enhanced Sertoli cell proliferation and differentiation while inhibiting WNT4-FOXL2 signaling. This was postulated to be mediated by FGF9-FGFR2c interactions. However, the study found species-specific regulation where the mouse SRY-SOX9-FGF9-FGFR2c positive feedback loop was dependent on SF1, whereas, in humans, it was SF1-independent [224]. Ulu et al. (2017) reported that FGF9-FGFR2c-mediated regulation of male differentiation is dose-dependent, where primordial germ cells exposed to low levels of FGF9 (0.2 ng/mL) stimulated p38-MAPK signaling that differentiated mouse primordial germ cells into pre-Sertoli cells, where high FGF9 levels (25 ng/mL) induced ERK1/2 signaling, promoting primordial germ cell proliferation and hinting at stage-specific FGF9 expression [227]. Further, in post-natal murine Leydig cells, FGF-mediated stimulation of testosterone production was observed via FGF9-Ras-MAPK, FGF9-PI3K, and PKA pathways. The occurrence of multiple FGFR isoforms (FGFR2c, 3c, and 4) and multiple signaling pathways suggests that the FGF9-mediated steroidogenesis in Leydig cells could be spatiotemporally regulated in a dose-dependent fashion [228]. Taken together, it is fair to conclude that FGF9 is a major determinant of male sex fate with a complex regulatory mechanism.

Contrary to widespread recognition of FGF9 for its male sex bias, Drummond et al. (2007) reported that FGF9 plays a role in progesterone production from ovaries in rats (inferred to be mediated via FGFR2c and FGF3c) [229] and in theca cell proliferation in cattle [230].

FGF9 is associated with Schwann cell transformation in mice [225] and in vitro (via the FGF9-FGFR2c-Akt pathway) during peripheral nerve regeneration [231]. Moreover, FGF9 is reported to be essential for the survival of GABAergic neurons in mice (via FGF9-adenylate cyclase-cAMP and ERK signaling) [232], dopaminergic neurons [233], and the development and function of Purkinje cells (via modulating neurotransmitter contents through ERK signaling) [234]. The importance of FGF9 in joint development (via FGF9-Erk1/2 signaling) [235], cleft palate formation during mouse embryogenesis (via FGF9-Wnt-β-catenin-TCF7L2 signaling) [236], osteogenic differentiation in mice (Akt signaling in osteoblasts and bone marrow mesenchymal stem cells) [237,238], and skeletal homeostasis [239] are also mentioned in the literature.

##### FGF16

The structure of FGF16 has not been resolved yet. However, based on its sequence similarity with the other members in the FGF9 subfamily, it can be deduced that FGF16 also contains a β-trefoil structure [240]. The secretion of FGF16 from cells through retrograde Golgi transport, where residues at the N-terminus and hydrophobic core are crucial for effective secretion [241]. FGF16 was reported to interact with FGFR2c and FGFR3c (highest preference) but not with FGFR1 and FGFR4 isoforms [242].

In the adult heart, endocardial cardiomyocyte proliferation was regulated by Foxp1-mediated suppression of Sox17, which led to the activation of FGF16 (and FGF20) signaling while repressing Wnt/β-catenin, whereas in the myocardium, the cardiomyocyte proliferation was mainly mediated by Nkx2.5 [243] and NF-κβ activation [244], suggesting a complex regulatory network for cardiomyocyte proliferation that is stage- and cell lineage-specific.

FGF16 is associated with inner ear development in chickens, which mediates proper orientation of the otic axis (anterior-posterior) [245,246], forebrain development of zebrafish (through facilitating GABAergic neuron differentiation via Hedgehog signaling) [247], activation of mesoderm during *Xenopus* larval development via activating the FGF9-MAPK pathway and sp5 and sp51 transcription activation [248], and zebrafish pectoral fin bud formation [249]. In Nile tilapia (*Oreochromis niloticus*), early phases of oocyte development [250], sex change in ricefield eels (*Monopterus albus*), and rat Leydig stem cell proliferation (via FGF9-PI3K-Akt1/2 and ERK1/2 pathways) were regulated by FGF16. Moreover, it plays a critical role in palatal closure in mice [251], murine tongue development during embryogenesis [252], and molar tooth development [253].

##### FGF20

The structure of FGF20 was published in 2009 (PDB ID—3F1R). Similar to other members in the FGF family, FGF20 also has a β-trefoil core flanked by flexible N- and C- termini helices [254]. The homodimerization-mediated autoinhibition was also present in FGF20, where N- and C-termini and core regions interact to form homodimers. The Trp147 residue in the β8-β9 turn of one FGF20 molecule forms critical interactions with Tyr70 and Arg193 in the other monomer to form a homodimer. As in FGF9, the FGF20 homodimer cannot interact with FGFR due to occlusion of receptor binding sites during dimerization. FGF20 acts in a paracrine fashion by interacting with FGFRs and HS. However, the dimeric form binds to HS much more strongly than the monomeric form, suggesting that the rate of diffusion and FGF gradient will be dependent on FGF-Hs interactions. This could be important for exerting cell-type-specific divergent roles [254].

The importance of FGF20 in post-natal heart development and homeostasis was reported in Cohen et al. (2007). The outflow tract and right ventricle of the heart are composed of anterior heart field progenitor cells, which are regulated by the Isl-1 transcription factor (hence, known as Isl-1 positive cardiac progenitor cells). The proper functioning of Isl-1-positive cardiac progenitor cells is regulated via the Wnt/β-catenin pathway and FGF signaling. FGF20 has been identified as a major player in this Wnt-FGF axis in Isl-1-positive cardiac progenitor cell proliferation, underscoring the importance of FGF20 in cardiac homeostasis and regeneration (together with FGF3, 10, and 16) [226]. FGF20 promoted cochlear and organ of Corti development by interacting with the FGFR1 isoform and transcription factor Sox2 [255] through MAPK and PI3K pathways [256].

### 3.2. Endocrine FGFs

#### FGF19 Subfamily

The endocrine FGF subfamily is composed of FGF19 (FGF15 in mice), FGF21, and FGF23 [5]. These proteins have a very low affinity for heparan sulfate and can join into the systemic circulation. They are known as FGF hormones or endocrine FGFs [5]. They regulate bile acid synthesis, glucose and lipid metabolism, and mineral metabolism [5,257,258]. The key difference of this subfamily is their dependence on klotho proteins as coreceptors during signaling. FGF19 and FGF21 depend on β-klotho for their intracellular signaling, whereas FGF23 interacts with α-klotho [259].

##### FGF19

FGF19 is the human ortholog of murine FGF15, with 51% sequence identity. The pair is often referred to as FGF15/19. In adults, FGF19 is present in the liver, the gallbladder epithelium, and the intestine, where it is primarily produced [260]. The gene encoding for FGF19 is located on chromosome 11 (11q13.1), and the full-length protein contains 216 amino acids with a hydrophobic region serving as signaling peptides that are cleaved after G22 in the mature protein [260,261]. The crystal structure of FGF19 revealed a β-trefoil core, similar to canonical FGFs [262,263]. Yet major differences are observed in essentially the heparin-binding sites. FGF19 has the longest β1-β2 loop among the entire family. The flexible structure is stabilized by a disulfide bond between C58 and C70, linking strands 2 and 3 [262]. The loop between strands 7 and 8 is a residue longer, changing the usual 3_10_ helix to an α-helix, which is linked to strand 6 through a second disulfide bond between C102 and C120 [262]. Another significant difference lies between the β10 and β12 regions. In place of a β11 strand, an α-helix (α11) is observed between K149 and K155 [263]. Both the α11 and the β1-β2 regions correspond to the primary and secondary heparin-binding sites in canonical FGFs, which may explain the reduced affinity of FGF19 for heparin [260,262,263].

Besides its role in the regulation of bile acid synthesis (Figure 2), FGF19 is also involved in maintaining lipid and glucose homeostasis [264,265,266]. Although the regulation of bile acids seems to be dependent on FGF19 binding to its receptor FGFR4, its metabolic effects would only require its coreceptor βklotho [267]. The main binding site of FGF19 with FGFR4 is located at the N-terminal between β1 and β2. The β1 strand in FGF19 is two residues longer, while the βC’—βE loop in FGFR4 is two residues longer than that of FGFR1c and FGFR2c [262,268]. These characteristics seem to favor a lower conformation between the two molecules, explaining the specific affinity of FGF19 for FGFR4 compared to the other receptors [262]. Removing residues 1–24 of the N-terminal of FGF19 and replacing them with that of FGF21 completely suppressed the ability to activate FGFR4 [267]. The low affinity of FGF19 for heparin is compensated by its binding with βklotho, which is a transmembrane protein primarily expressed in the liver, the pancreas, and the adipocytes [268,269]. Wu et al. have also demonstrated that heparin may be necessary for stronger binding of FGF19 with βklotho [270]. FGF19 was suggested by Lee et al. to interact with βklotho in a manner similar to FGF21 [271]. Glu693 in the D2 domain of the extracellular region of βklotho interacts with Ser211-Pro212-Ser213 residues of FGF19, while interactions with the D1 domain are insured by Asp198-Pro199 of FGF19 [271]. 

FGF19 plays a key role in regulating glucose homeostasis independently of insulin by promoting glycogen synthesis via the inactivation of glycogen synthase kinase 3α and 3β. It also lowers blood glucose levels by inhibiting the activity of the transcription factor CREB, which controls the genes involved in gluconeogenesis [272]. In murine models, FGF19 enhances energy expenditure through increased hepatic lipid oxidation and the formation of brown adipose tissue (BAT) [272]. Plasma levels of FGF19 peak around two hours post-meal, but its metabolic effects can take days to be noticeable [273]. The actions of FGF19 are mediated through the FGFR/β-Klotho–ERK–RSK pathway [272]. However, at elevated levels, FGF19 has been implicated in hepatocellular carcinoma, particularly through signaling involving FGFR4 [274]. To address this issue, Aldafermin (NGM282), an FGF19 analog in clinical trials, was engineered with a modified N-terminal to reduce its affinity for FGFR4, thereby minimizing the risk of liver cancer while preserving its metabolic benefits [275].

##### FGF21

FGF21 is the second member of the endocrine FGF subfamily. The mature peptide is composed of 209 amino acid residues. The transcription of FGF21 is differentially regulated in a cell-type-dependent manner [273]. In the liver, peroxisome proliferator-activated receptor α (PPARα) and cAMP response element-binding protein (CREB) induce *Fgf21* gene transcription upon fasting or ketone-rich diet, whereas in the adipocytes, PPARγ is the main mediator of *Fgf21* transcription [258,276,277]. The crystal structure of FGF21 exhibits shorter, less compact (not fully closed) β-trefoil folds due to unusual stretch between β strands 10 and 12, which houses a major heparin binding site on paracrine FGFs. Furthermore, FGF21 (also FGF19 and 23) lacks the β 11 strand as well [271]. The less compact structure of endocrine FGFs makes them more prone to thermal and proteolytic degradation, hence shorter half-lives in vivo. The C-tail of FGF21 interacts with β-klotho [271]. The structure elucidated the presence of a much more rigid C-terminus of FGF21 than its FGF19 counterpart due to the extensive network of hydrophobic interactions. The Asp192 and Pro193 (D-P motif) and Ser204, Pro205, and Ser206 (S-P-S motif) in the FGF21 C-terminus are critical residues that interact with β-klotho [271]. These residues are also conserved in FGF19; however, they are not conserved in FGF23.

The main functions of FGF21 include nutrient and energy homeostasis. More specifically, FGF21 is associated with lipid oxidation, enhancement of insulin sensitivity, promotion of energy expenditure, and regulation of macronutrient preference [271]. The FGF21-FGFR1c-β-klotho complex is essential for downstream signaling. Further, it is associated with cardio protection [278], myoblast differentiation [279], and sweet and alcohol preference [280]. Further, dysregulated FGF21 levels are affiliated with certain metabolic diseases such as liver diseases (e.g., nonalcoholic fatty liver disease [281] and nonalcoholic steatohepatitis [282], liver fibrosis [283]), obesity [282], diabetes [283], cardiovascular diseases (e.g., atherosclerosis [284]), and cancer [276].

In white adipose tissues (involved in lipid storage), FGF21 induced uncoupling protein 1 (UCP1) expression via PGC1α, promoting thermogenesis. However, during lipolysis, PPARα induces CPT1a and HMGCS2 gene transcription (involved in fatty acid oxidation). Concurrently, PPARα induces FGF21 as well; hence, the combinatory effect of all these enhances lipolysis in white adipose tissues and ketogenesis in the liver [285]. During energy metabolism, FGF21 in adipocytes activates AMPK-SIRT1-PGC1α signaling, leading to enhanced mitochondrial oxidation [286]. Moreover, FGF21 can induce gluconeogenesis in the liver by interacting with the brain and regulating glucose homeostasis. PPARα induces FGF21 production from the liver during prolonged starvation, can be circulated into the brain, and activates the hypothalamic-pituitary-adrenal axis in order to secrete corticosterone to induce gluconeogenesis in the liver. In hypothalamic neurons, FGF21-FGFR1c-mediated ERK 1/2 was reported to coordinate the signaling pathway [287].

In nonalcoholic fatty liver disease, hepatic steatosis and inflammation led to reduced FGF21 levels due to NF-κB-mediated *Fgf21* transcription or interaction with SREBP1c with FGF21 [281]. In the heart, PPARα and Sirt1 regulated FGF21 transcription and conferred protection against cardiac hypertrophy by FGF21-mediated amelioration of oxidative stress and inflammation [278]. During myogenesis, FGF21-SIRT1-AMPK-PGC1α signaling induced a metabolic change where myofibers switched from anaerobic respiration to aerobic respiration, facilitating myogenic differentiation [279].

##### FGF23

The *Fgf23* gene is located on chromosome 12p13.32. The immature transcript of FGF23 contains 251 amino acids with a molecular mass of 32 kDa [5]. During maturation, the N-terminal signaling peptide (24 amino acid residues) will be cleaved, giving rise to the biologically active form of FGF23 (227 amino acids; intact FGF23 or iFGF23), and the mature peptide hormone is released into the circulation [288]. O-glycosylation at Thr178 and phosphorylation at Ser180 stabilize the FGF23 protein [288]. The interactions between FGF23 and its specific coreceptor, alpha klotho (aKL), with FGFR will activate downstream signaling pathways to exert its biological functions. In the absence of aKL, FGF23 shows a very low affinity for FGFR, underscoring the importance of aKL in FGF23-mediated signaling [289]. Even though FGF23 can act as a hormone that can join systemic circulation, the restricted expression of aKL at the kidneys confers FGF23-FGFR-aKL signaling to occur only at target sites [5,290]. FGF23 prefers to bind to FGFR1c. However, the literature has also reported FGF23 interactions with FGFR3 and FGFR4 [290]. FGF23 has the longest C terminus (contains 72 amino acids) of all endocrine FGFs and is postulated to contain two aKL binding sites, unlike FGF19 and FGF21, which only have one binding surface for beta klotho (bKL) [291]. The RXXR cleavage site at the boundary of the C tail is critical to maintaining biologically active intact FGF23 (iFGF23) levels in the body where proteolytic cleavage at the RXXR site frees the C terminus (cFGF23) from the iFGF23 [288]. The cFGF23 fragment is postulated to exert an autoinhibitory function that antagonizes the function of iFGF23. In autosomal dominant hypophosphatemic rickets (ADHR), a mutation at the cleavage site abolishes cleavage of FGF23, resulting in a higher concentration of iFGF23 in the body, leading to phosphate wasting [292].

Higher plasma FGF23 levels are associated with phosphate-wasting disorders such as autosomal dominant hypophosphatemic rickets (ADHR) [292], bone cancers (e.g., tumor-induced osteomalacia (TIO)) [293], kidney disorders (e.g., chronic kidney disease, acute kidney injury, diabetic nephropathy) [290], heart conditions (coronary heart disease, atherosclerosis, ischemia-reperfusion), and inflammation [5].

The key biological function of FGF23 (canonical FGF23 signaling) (Figure 3a), phosphate homeostasis, is mediated by reducing the surface expression of sodium-phosphate co-transporters NPT2a and NPT2c, driven by FGF23-FGFR1c-aKL-ERK1/2 and FGF23-FGFR1c-aKL-SGK1 signaling leading to phosphorylation of NHERF1 [290,294,295]. This results in the internalization of NPT2a and NPT2c into proximal tubules and subsequent degradation. On the other hand, FGF23-mediated phosphorylation of ERK1/2 facilitates the colocalization of aKL and TRVP5 calcium channels in the renal distal tubules, leading to increased calcium reabsorption (decreased calcium excretion) in wild-type mice [296]. Moreover, FGF23 can inhibit osteocyte differentiation via PI3K-Akt-mTOR signaling [297].

Conversely, non-canonical FGF23 signaling (Figure 3b) is mediated through FGFR3 and FGFR4 isoforms independent of aKL [290]. During left ventricular hypertrophy and myocardial fibrosis, FGF23 interaction with FGFR4 gives rise to the PLCγ-calcineurin-NFAT3 signaling pathway, aggravating the disease condition [298]. Moreover, in osteoblasts, FGF23 can inhibit non-tissue-specific alkaline phosphatase (TNAP) activity through FGF23-FGFR3-ERK1/2 signaling in the absence of aKL, promoting dysregulated bone mineralization [299,300]. In rheumatoid arthritis, FGF23 enhanced a proinflammatory cytokine, IL-1β, in synovial fibroblasts through FGF23-FGFR1c-mediated PI3K-Akt and NF-κB pathways promoting joint pain, swelling, and bone deformities [301]. On a positive note, FGF23 was reported to be involved in alleviating lung ischemia-reperfusion injury, where FGF23-FGFR4-ERK1/2 signaling mediated activation of erythropoietin. The erythropoietin upregulation attenuated apoptosis and tissue damage in the lungs, improving lung ischemia-reperfusion injury [302] (Figure 4).

**Figure 3 biomolecules-14-01622-f003:**
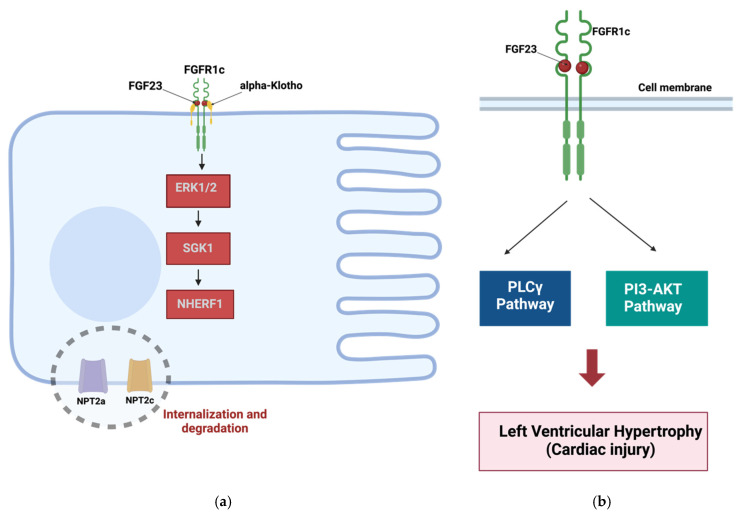
Canonical and non-canonical functions of FGF23. (**a**) Canonical FGF23 signaling where FGF23-FGFR1c-alpha klotho interaction leads to internalization of NPT2a and NPT2c into proximal tubules and subsequent degradation, reducing their surface abundance. This reduces the phosphate reabsorption from proximal tubules and increases phosphate wasting. (**b**) Non-canonical FGF23 signaling is mediated through FGFR3 and FGFR4 isoforms independent of alpha klotho. During left ventricular hypertrophy and myocardial fibrosis, FGF23 interaction with FGFR4 gives rise to the PLCγ-calcineurin-NFAT3 and PI3K-AKT signaling pathways, aggravating the disease condition. This image was created with Biorender (https://biorender.com).

**Figure 4 biomolecules-14-01622-f004:**
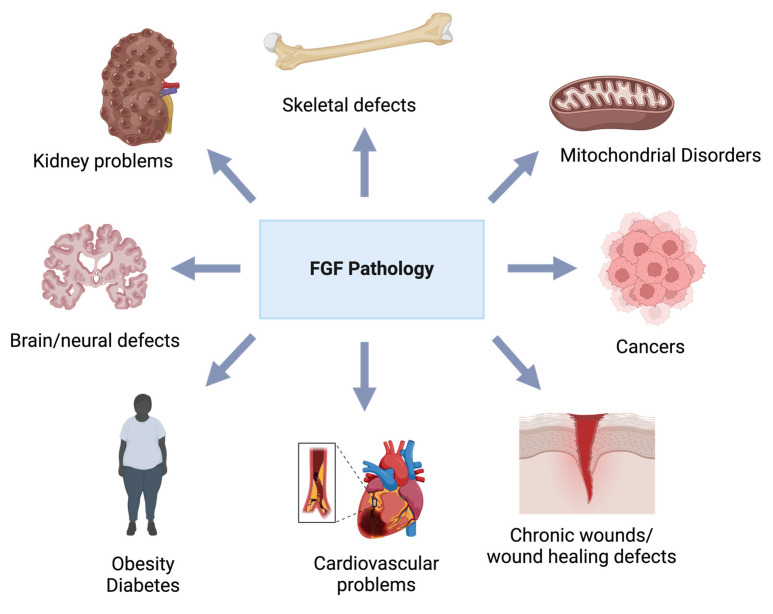
FGF-related pathological conditions. Due to dysregulated FGF signaling, many diseases can arise, such as skeletal defects, mitochondrial disorders, cancer, kidney diseases, obesity, dermatological issues, cardiovascular complications, and brain defects.

**Table 1 biomolecules-14-01622-t001:** Summary of FGF-mediated physiological functions and disorders.

Subfamily	FGF	Main Functions	Diseases
FGF1	FGF1	Cell proliferation and, differentiation Angiogenesis [13] Wound healing [23]	Cancer (pancreatic cancer [34], non-small lung cancer [32], gastric cancer [303], breast cancer [31], colorectal cancer [33], ovarian cancer [31])
	FGF2	Angiogenesis [64] Wound healing [64] Cartilage and tendon repair [304]	Osteoarthritis [56] Cancer (non-small cell lung cancer [305], squamous lung cancer [306], hepatocellular carcinoma [307])
FGF4	FGF4	Limb development [308] Wound healing [167] Maintenance of pluripotency [165,309] macrophage survival [166].	Liver diseases (Non-alcoholic fatty liver disease, autoimmune hepatitis) [166]
	FGF5	Hair growth regulation [168]	Lung cancer [310]
	FGF6	Angiogenesis [311] cardiac repair [312]	Bladder cancer [311]
FGF7	FGF3	Ear [313], Brain [76], Cardiovascular [314] and Head skeleton development [315]	Auditory impairments/ear malformations [316] Cancer (Kaposi’s sarcoma [82], hepatocellular carcinoma [83], non-small cell lung disease [86])
	FGF7	Wound healing [317,318] Hair growth [319] Glandular development (mammary gland [95], submandibular glands [102])	Mandible defects [320] Cancer (leukemia [108], gastric cancer [110])
	FGF10	Glandular development (mammary glands [321], salivary glands [116]), Lung [118], Neural development [119]	Lung and limb agenesis [322] Glandular defects (e.g., Lacrimo-auriculo-dento-digital syndrome [118], aplasia of the lacrimal and salivary glands [118])
	FGF22	Brain [323] and ear development [324] Spinal cord remodeling [141] Liver regeneration [140]	Auditory defects/deafness [324]. Cancer (ovarian cancer [325], pancreatic cancer [326], lung cancer [327])
FGF8	FGF8	Limb development [328] Brain development [190,329]	Neural defects [330]
	FGF17	Brian development [329,331]	Cognitive impairments [332,333] Cancer (breast cancer and ovarian cancer [334])
	FGF18	Brian development [335] Heart and cartilage development [335]	Osteoarthritis [336,337,338,339]. Cancer (colorectal cancer, ovarian cancer, breast cancer [335])
FGF9	FGF9	Male sex development [340] Neural differentiation [225]	Sertoli cell-only syndrome [341] Huntington’s disease [342,343] Cancer (Prostate [344,345], gastric cancers [346], Leydig tumor [347,348], Hepatocellular carcinoma [349,350])
	FGF16	Embryonic heart development [351,352,353] Inner ear development [245,246] Brain development [247] Oocyte development [250]	Cancer (Breast [242] and Lung cancer [354], Hepatocellular carcinoma [355])
	FGF20	Heart development [356], Cochlear and organ of Corti [357] Cardiac and kidney regeneration [226,358]	Deafness [359]. Parkinson’s disease [262,360]
FGF19	FGF19	Regulation of bile acid synthesis [361] Glucose and lipid homeostasis [267,362]	Bile acid disorders [260,363] Non-alcoholic fatty liver disease [266,273] Non-alcoholic steatohepatitis [266,273] Type 2 diabetes [273] Obesity [266,273]
	FGF21	Glucose and lipid homeostasis [5]	Type 2 diabetes [364] Non-alcoholic fatty liver disease [5] and Non-alcoholic steatohepatitis [5] Obesity [5], Cancer (breast cancer, thyroid cancer, renal cancer, endometrial cancer) [5]
	FGF23	Vitamin D and phosphate level homeostasis	Tumor-induced osteomalacia [1] X-linked hypophosphatemia [1] kidney diseases, Left ventricular hypertrophy [1]

## 4. Conclusions

FGF-FGFR signaling is crucial in development and pathogenesis. An in-depth understanding of interactions in FGF-mediated signaling pathways with other signaling pathways, genetic regulation of FGFs, and associated molecular mechanisms will be necessary to implement new approaches that are more targeted to a plethora of FGF-mediated genetic, metabolic, and degenerative disorders, cancer, and wound healing. Despite the progress made, challenges remain in fully understanding the spatial and temporal regulation of FGF signaling; hence, it warrants future studies to unravel the FGF-FGFR precise signaling in cellular and disease progression. In this review, we provide a holistic view of differential signaling pathways FGFs initiate under different contexts. Further, we have documented pathological conditions that arise due to FGF-FGFR signaling dysregulation.

## Figures and Tables

**Figure 1 biomolecules-14-01622-f001:**
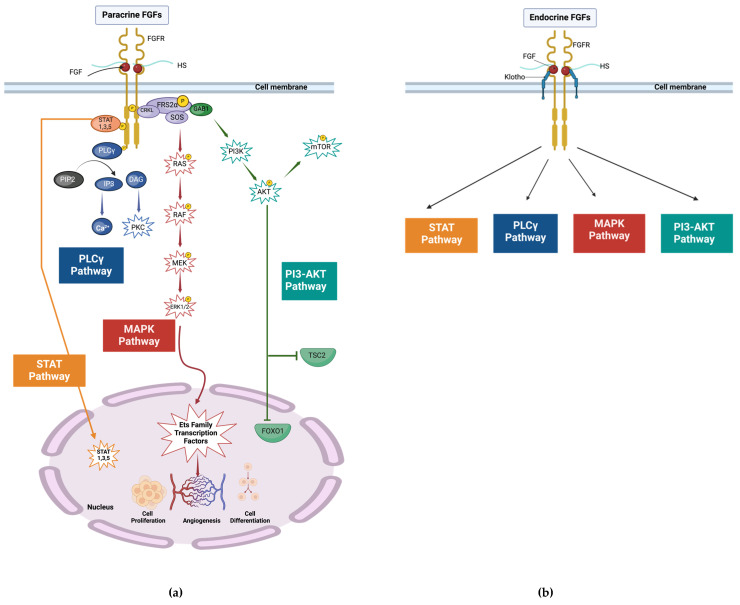
FGF signaling pathways. (**a**) Paracrine FGF signaling—The formation of the FGF-FGFR-HS complex induces 4 major FGFR-mediated intracellular signaling pathways: RAS-MAPK, PI3K-AKT, PLCγ, and STAT pathways. The RAS-MAPK pathway begins when FRS2α interacts with CRKL and stimulates phosphorylation of FRS2α by activated FGFR kinase. The adapter protein GRB2 is subsequently recruited by activated FRS2α, which then leads to recruitment of SOS. SOS activates RAS GTPases associated with the MAPK pathway. MAPK pathway-induced Ets family transcription factors, which is a family of transcription factors associated with inducing cell proliferation, angiogenesis, and cell differentiation; classic responses of paracrine FGFs (the intracellular mediators associated with MAPK pathway are shown in red). Alternatively, the FRS2α-GRB2 activated complex can recruit another adapter protein, GAB1, leading to the activation of the PI3K enzyme. PI3K then phosphorylated another enzyme called AKT. The physiological roles of AKT include activation of mTOR complex 1 (TSC2 and FOXO1 will be inhibited during AKT activation) and the ensuing PI3-AKT pathway (shown in green). The PLCγ pathway (in blue) is initiated when activated FGFR recruits the PLCγ enzyme, which induces hydrolysis of PIP_2_ into IP_3_ and DAG. IP_3_-mediated signaling leads to calcium ion release from intracellular storages and activation of subsequent signaling pathways, whereas DAG stimulates PKC-mediated downstream signaling. Further, FGF-FGFR-mediated activation of the STAT pathway (in orange) is initiated by phosphorylation of STAT1, 3, and 5 in the cytoplasm, which then translocate to the nucleus. STATs act as transcription factors that are involved in the regulation of FGF-mediated signaling. (**b**) Endocrine FGF signaling complexes are comprised of klotho coreceptors in addition to the canonical FGF-FGFR-HS complex. FGF19 and FGF21 form stable complexes with β-klotho to initiate intracellular signaling pathways, whereas FGF23 signaling is mediated by α-klotho. This image was created with Biorender (https://biorender.com/).

**Figure 2 biomolecules-14-01622-f002:**
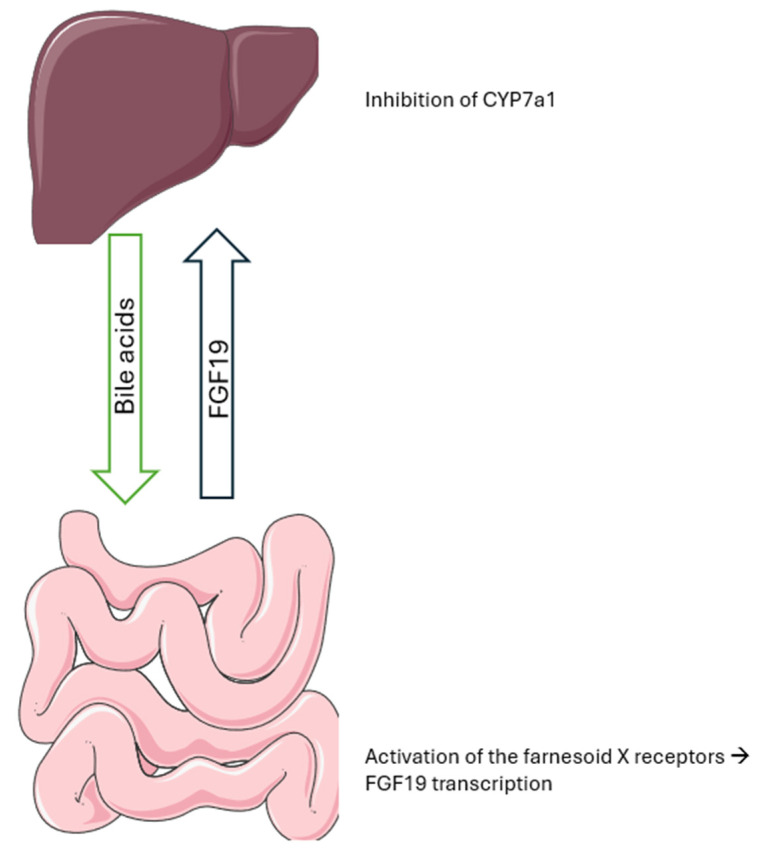
After meals, bile acids travel to the intestine, where they activate the farnesoid X receptors, which in turn signal the transcription of FGF19 in the ileum. The mature FGF19 then travels through the hepatic portal to the liver, where it inhibits bile acid synthesis through a feedback mechanism.

## Data Availability

No new data were created or analyzed in this review. Data sharing is not applicable to this article.

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
