# Peer review of "Decoding FGF/FGFR Signaling: Insights into Biological Functions and Disease Relevance"

_biomolecules, 2024, doi:10.3390/biom14121622_

Round 1

Reviewer 1 Report

Comments and Suggestions for Authors

Thank you for the opportunity to review your manuscript entitled "Decoding FGF/FGFR Signaling: Insights into the Link Between Biological Function and Disease." Your work provides a comprehensive overview of the FGF signaling pathways and their implications in disease states, which is a valuable contribution to the field.

General Comments:
While the manuscript covers a wide range of FGFs and their associated signaling pathways, we believe the discussion could benefit from a deeper exploration of specific aspects. Currently, the discussion is somewhat broad and lacks sufficient depth in some areas. We understand that focusing on all FGFs presents challenges in maintaining coherence. Nevertheless, a more focused approach in certain sections could enhance the clarity and impact of your arguments.

Specific Comments:
Figures: The figures are well-designed, straightforward, and effectively support the manuscript’s content. No revisions are necessary in this regard.
Scope and Depth: If possible, consider narrowing the focus to key FGFs or signaling pathways that are most critical to understanding the connection between biological function and disease. This could allow for a more in-depth analysis without sacrificing clarity.
Recommendation:
Despite the noted limitations in depth, we believe the manuscript makes a significant contribution to the understanding of FGF-mediated signaling pathways and their role in disease. Therefore, we recommend proceeding with the manuscript in its current form, with minor adjustments to improve coherence and depth where feasible.

Thank you again for sharing your work. We look forward to seeing its final version.

Author Response

“While the manuscript covers a wide range of FGFs and their associated signaling pathways, we believe the discussion could benefit from a deeper exploration of specific aspects. Currently, the discussion is somewhat broad and lacks sufficient depth in some areas. We understand that focusing on all FGFs presents challenges in maintaining coherence. Nevertheless, a more focused approach in certain sections could enhance the clarity and impact of your arguments.”

“Scope and Depth: If possible, consider narrowing the focus to key FGFs or signaling pathways that are most critical to understanding the connection between biological function and disease. This could allow for a more in-depth analysis without sacrificing clarity.”

Response: We thank the reviewer for the suggestion on our manuscript. In our contribution, we want to provide holistic view of different signaling pathways mediated by FGFs. In this context, we believe that our review article provides information on the recent cellular and therapeutic roles of both mitogenic and metabolic FGFs. We hope our comprehensive review will meet the criteria of your prestigious journal such as Biomolecules.      

Reviewer 2 Report

Comments and Suggestions for Authors

The article "Decoding FGF/FGFR Signaling: Insights into Biological Functions and Disease Relevance" elaborates the review of the role of fibroblast growth factors (FGFs) and the corresponding receptors, FGFRs, in various biological functions and disease pathogenesis. FGF/FGFR proteins carry out important cellular processes such as proliferation, differentiation, tissue repair, and metabolic regulation. Besides, further details of FGF-FGFR signaling pathways and their crosstalk with other signaling networks are discussed at a high level in this review. It also emphasizes the role of these molecules in development, metabolism, and homeostasis. Finally, the review tries to expand this understanding of FGF-FGFR signaling to devise therapeutic interventions for pathological conditions associated with FGFs.

In my opinion the article is quite important because it expands the knowledge about FGF/FGFR pathways, which play a crucial role in fundamental biological processes and the pathophysiology of many diseases.

In my opinion, the following few key issues could be worked upon, and if that is done, it should be publishable.

FGF subfamily separation: The addition of subheadings for each FGF subfamily (e.g., FGF1, FGF4) would make this article clearer and more organized. The reader would better be able to follow the role of each subfamily and its interactions with the FGFRs.

Explanation of Terminology: Signaling through pathways such as MAPK or PI3K-AKT may not be understandable to all readers. Explanations in plain language or footnotes with definitions would help in easier understanding.

Avoid Repetitions: In several places the function of specific pathways or the mechanism of action of some subfamilies of FGFs is repeated. These can be organized into central sections, for greater coherence and clarity.

Summary: Add a concluding chapter that summarizes the effects of FGFs in different tissues and pathologies, thereby providing a coherent view of their importance.

Recommendations for Future Research: Extensions directed towards specific areas where research is lacking or focusing on certain applications of FGF-FGFR signaling, such as targeted therapies in light of reduced side effects.

Action Graphs by Tissue and Condition: A visual showing how FGF-FGFR interactions affect various tissues or conditions could add value, helping the reader compare their effect in different contexts.

References: The references should be done in the style of the journal.

Author Response

“FGF subfamily separation: The addition of subheadings for each FGF subfamily (e.g., FGF1, FGF4) would make this article clearer and more organized. The reader would better be able to follow the role of each subfamily and its interactions with the FGFRs.”

Response:  We thank the reviewer for the suggestion on our manuscript and we accepted the reviewer’s comment and added the subheadings for each FGF subfamily.

“Explanation of Terminology: Signaling through pathways such as MAPK or PI3K-AKT may not be understandable to all readers. Explanations in plain language or footnotes with definitions would help in easier understanding.”

Response: We thank the reviewer for the suggestion on our manuscript. We have outlined the major signaling pathways that FGF-FGFRs initiate in our Figure 2. For more clarity we have included a list of abbreviations for each acronym in these pathways for easier understanding of the readers (from Line 920).

“Avoid Repetitions: In several places the function of specific pathways or the mechanism of action of some subfamilies of FGFs is repeated. These can be organized into central sections, for greater coherence and clarity.”

Response: We thank the reviewer for the suggestion on our manuscript. We have edited the manuscript accordingly to avoid repetition.

“Summary: Add a concluding chapter that summarizes the effects of FGFs in different tissues and pathologies, thereby providing a coherent view of their importance.”

Response: We thank the reviewer for the suggestion on our manuscript. We added a revised conclusion with more details than provided in the initial submission (line 901 to 905).

“Recommendations for Future Research: Extensions directed towards specific areas where research is lacking or focusing on certain applications of FGF-FGFR signaling, such as targeted therapies in light of reduced side effects.”

Response: We thank the reviewer for the suggestion on our manuscript. We believe reviewer’s suggestion warrants much detailed explanations which is beyond the scope of our current manuscript. Our goal for this publication was to highlight the complexity and diversity of FGF-FGFR signaling.

“Action Graphs by Tissue and Condition: A visual showing how FGF-FGFR interactions affect various tissues or conditions could add value, helping the reader compare their effect in different contexts.”

Response: We thank the reviewer for the suggestion on our manuscript. We have already included selected examples of canonical and non-canonical FGF-FGFR pathways in Figures 2 and 3.

“References: The references should be done in the style of the journal.”

Response: We thank the reviewer for the suggestion on our manuscript. We used the MDPI reference style recommended by the journal.

Reviewer 3 Report

Comments and Suggestions for Authors

Edirisinghe et al wrote an extremely extensive review focusing at the different FGF factors. The function and differences of the FGF receptors are given little space and importance. Nonetheless, they are somehow mentioned as interaction partners of the FGFs. Unfortunately, the review has hardly any structure. The presentation of some table would be highly recommended. For example, it would make sense to summarize the structural features of the different FGFs in a clearly arranged table to highlight the similarities and differences obvious. In another table, the authors could summarize the knock-out phenotypes of the FGF or the diseases they are involved in. In addition, a short chapter on the selection of the literature cited should be included. Furthermore, the authors should summarize what new features their work has compared to the existing works and provide  some less general but new conclusions. Figure 1 is not giving a lot of information and Figure2 and Figure 3 are repetitions of already well-known illustrations.

Minor Points

Lines 36 to 41 say the same as 44-48

The review is supposedly divided into 5 sections. However, there is only Chapter1 Introduction and Chapter5 Conclusion. Chapter 2, 3 and 4 are not mentioned.

Author Response

“Lines 36 to 41 say the same as 44-48”

Response: We thank the reviewer for the feedback on our manuscript. We have omitted the repetition (Lane 44 to 46).

“The review is supposedly divided into 5 sections. However, there is only Chapter1 Introduction and Chapter5 Conclusion. Chapter 2, 3 and 4 are not mentioned.”

Response: We thank the reviewer for the feedback on our manuscript. Since this is a review, we cannot subdivide our manuscript into typical chapter format given by Biomolecules journal which is Chapter 2,3 and 4 being Materials and Methods, Results and Discussion respectively. However, we have systematically numbered each subfamily and subfamily members for more clarity.

“Edirisinghe et al wrote an extremely extensive review focusing at the different FGF factors. The function and differences of the FGF receptors are given little space and importance. Nonetheless, they are somehow mentioned as interaction partners of the FGFs. Unfortunately, the review has hardly any structure. The presentation of some table would be highly recommended. For example, it would make sense to summarize the structural features of the different FGFs in a clearly arranged table to highlight the similarities and differences obvious. In another table, the authors could summarize the knock-out phenotypes of the FGF or the diseases they are involved in. In addition, a short chapter on the selection of the literature cited should be included. Furthermore, the authors should summarize what new features their work has compared to the existing works and provide some less general but new conclusions. Figure 1 is not giving a lot of information and Figure2 and Figure 3 are repetitions of already well-known illustrations.”

Response: We thank the reviewer for the feedback on our manuscript. We believe reviewer’s suggestion warrants much detailed explanations which is beyond the scope of our current manuscript. Our goal for this publication was to highlight the complexity and diversity of FGF-FGFR signaling.

Round 2

Reviewer 2 Report

Comments and Suggestions for Authors

The authors made all corrections. The revised version is completely agreeable to me, for this reason, I recommend the publication of the manuscript.

Author Response

The authors made all corrections. The revised version is completely agreeable to me, for this reason, I recommend the publication of the manuscript.

Response: We thank the reviewer for their valuable and thoughtful comments on the previous version of our manuscript. Their comments have been very useful in improving the quality of the manuscript”.

Reviewer 3 Report

Comments and Suggestions for Authors

Since there are almost no changes compared to the earlier version. In just can repeat:

The function and differences of the FGF receptors are given little space and importance. Nonetheless, they are somehow mentioned as interaction partners of the FGFs. Unfortunately, the review has hardly any structure. The presentation of some table would be highly recommended. For example, it would make sense to summarize the structural features of the different FGFs in a clearly arranged table to highlight the similarities and differences obvious. In another table, the authors could summarize the knock-out phenotypes of the FGF or the diseases they are involved in. In addition, a short chapter on the selection of the literature cited should be included. Furthermore, the authors should summarize what new features their work has compared to the existing works and provide  some less general but new conclusions. 

Author Response

1. “The function and differences of the FGF receptors are given little space and importance. “

Response: We have included a brief explanation of FGFRs (Lane 56 to 79)

2. Unfortunately, the review has hardly any structure

Response: Please note that we have structured our manuscript in a numerical order where headings and subheadings were numbered accordingly

3. The presentation of some table would be highly recommended. For example, it would make sense to summarize the structural features of the different FGFs in a clearly arranged table to highlight the similarities and differences obvious. In another table, the authors could summarize the knock-out phenotypes of the FGF or the diseases they are involved in.

Response: We have included “Table 1 : Summary of FGF-mediated physiological functions and disorders” (Page 21 to 23)

4. In addition, a short chapter on the selection of the literature cited should be included.

Response: We acknowledge the reviewer’s suggestion, and we included the method for literature selection (Lane 52 to 55)

5. Furthermore, the authors should summarize what new features their work has compared to the existing works and provide some less general but new conclusions. 

Response: This is a review paper which summarizes current knowledge in already published literature. Hence, the reviewer’s suggestion to make new conclusions is not quite clear. However, with all due respect, we have rewritten the conclusion section honoring the reviewer’s comments (Lane 930 to 936)